# Stem-Cell Theory of Cancer: Implications for Antiaging and Anticancer Strategies

**DOI:** 10.3390/cancers14051338

**Published:** 2022-03-04

**Authors:** Shi-Ming Tu, Louis L. Pisters

**Affiliations:** 1Division of Hematology/Oncology, University of Arkansas for Medical Sciences, Little Rock, AR 72205, USA; 2Department of Urology, The University of Texas MD Anderson Cancer Center, Houston, TX 77030, USA; lpisters@mdanderson.org

**Keywords:** cancer stem cells, aging, microbiome, naked mole-rat, Down’s syndrome, progeria, antiaging

## Abstract

**Simple Summary:**

A stem-cell theory of cancer connects aging with cancer. It indicates that aging is a stemness process and cancer is a stem-cell disease. It implicates that a pertinent scientific strategy and proper research endeavor may provide us with realistic antiaging objectives and superior anticancer outcomes. In this perspective, we illustrate that a stem-cell origin of aging and cancer reiterates a fundamental oncological principle: although genetic makeup may be pivotal, cellular context is paramount. When the genome and epigenome that regulate aging and malignancy are also stemness genes and stem-like properties, they reaffirm the essential role stem-cell quality and quantity play in our lifespan and in the formation of cancer.

**Abstract:**

A stem-cell theory of cancer predicates that not only does the cell affect the niche, the niche also affects the cell. It implicates that even though genetic makeup may be supreme, cellular context is key. When we attempt to solve the mystery of a long cancer-free life, perhaps we need to search no further than the genetics and epigenetics of the naked mole-rat. When we try to unlock the secrets in the longevity and quality of life, perhaps we need to look no further than the lifestyle and habits of the super centenarians. We speculate that people with Down’s syndrome and progeria age faster but have fewer cancers, because they are depleted of stem cells, and, as a consequence, have fewer opportunities for stem cell defects that could predispose them to the development of cancer. We contemplate whether these incredible experiments of nature may provide irrefutable evidence that cancer is a stem-cell disease—fewer aberrant stem cells, fewer cancers; no defective stem cells, no cancer. In this perspective, we investigate a stem-cell origin of aging and cancer. We elaborate an intriguing inverse relationship between longevity and malignancy in the naked mole-rat, in Down’s syndrome, and in progeria. We postulate that stem-cell pools and stemness factors may affect aging and dictate cancer. We propose that a healthy microbiome may protect and preserve stem cell reserves and provide meaningful antiaging effects and anticancer benefits.

“*Old age isn’t so bad if you consider the alternative.*”—**Maurice Chevalier**

## 1. Introduction

How age connects with stem cells makes sense in real life, because stem cells regenerate and rejuvenate less well as we get older. When we age, the pool of stem cells becomes depleted and the stem cells lose their vigor and vitality.

How cancer connects with stem cells is also increasingly evident in the laboratory and the clinic. Stemness properties seem synonymous with malignant characteristics. Both cancer cells and stem cells mobilize and metastasize, differentiate and diversify. They are shielded from the severity of pharmaceutical agents and sheltered from the scrutiny of the immune system.

The stem-cell theory of cancer connects aging with cancer. It indicates that aging is a stemness process and cancer is a stem-cell disease. It implicates that a pertinent scientific strategy and proper research endeavor may provide realistic antiaging effects and superior anticancer outcomes.

In this perspective, we investigate a stem-cell origin of aging and cancer. We discuss an intriguing inverse relationship between longevity and malignancy in the naked mole-rat, in Down’s syndrome, and in progeria. We speculate that stem-cell pools and stemness factors may affect aging and dictate cancer. We propose that a healthy microbiome may protect stem cell reserves and provide healthful antiaging effects and meaningful anticancer benefits.

## 2. Age and Cancer

In scientific research, we need to be disciplined in our application of the scientific method. We need to make pertinent observations that enable and empower us to formulate credible theories and useful hypotheses. We need to test our hypotheses in the laboratory to help us understand the basic mechanisms of our observations, elucidate the origin and nature of their occurrence, and predict or produce something to benefit mankind.

Because older adults have a cancer rate 1000 times higher than young adults, we often hypothesize that the process of aging contributes to the formation of cancer. However, there is also evidence suggesting that age may have no direct or independent effect on the development of cancer [1]. According to in vivo studies in which carcinogenic treatments were started at different ages, cancer occurred at about the same rate in older as in younger animals [1]. In some experiments, cancer developed less rapidly in older compared with younger animals [2,3].

Association does not equal causation. Older people may have gray hair and wrinkled skin. However, gray hair and wrinkled skin are effects rather than the cause of aging.

The observation that old age is associated with cancer but does not necessarily cause cancer suggests that the accumulation of genetic mutations over time and with age does not account for the formation of cancer. This paradox of age and cancer is also evident in findings showing that species with vastly different lifespans have similar cancer risks [1]. Hence, a blue whale with a lifespan of about 80 years and a mouse with a lifespan of about 2.5 years have comparable lifelong risks of developing cancer.

Indeed, the Mammalian Methylation Consortium [4] found that an “epigenetic clock” of almost 800 methylation sites could reliably estimate an individual’s age relative to the maximum lifespan of its species (in all 192 mammalian species). Importantly, longer-lived mammals have more stable epigenetic marks, involving developmental genes such as HOX and PAX, that link growth with aging, as well as genes related to mitochondrial function, which is known to play a role in the aging process, suggesting that they have maintained a youthful genome—or more accurately, a youthful epigenome, i.e., with stemness marks.

## 3. Methuselah

Supposedly, Methuselah lived longer than any person ever recorded: 969 years (Genesis 5:27). Perhaps people in the Old Testament used a different estimation to tabulate age [5], or perhaps they were blessed with an incredibly youthful epigenome.

When lifespan-related methylation tends to be associated with genes related to development, it is a reminder that genes essential for regulating lifespan and aging are also stemness genes. It reaffirms the key role that the quality and quantity of stem cells play in the lifespan. It reinforces the biological precept that although genetic makeup may be important, cellular context is pivotal. It reiterates the fundamental principle that oncology recapitulates ontogeny.

Hence, animals with longevity have somehow managed to safeguard their stem cells. As a result, they are also spared of cancer, because they have better means of keeping their stem cells healthy and robust.

The naked mole-rat (*Heterocephalus glaber*) lives more than 35 years, whereas a house mouse lives only 2 to 3 years. Naked mole-rats have more accurate ribosomes, vigorous proteasomes, and potent DNA repair genes (such as Sirt6) that limit damages and ensure the integrity of their cellular proteins and DNA [6,7,8]. We surmise that when animals have special capabilities to shield their stem cells, they are blessed with longevity without malignancy.

Hadi et al. [9] demonstrated that genes known to cause cancer in mouse and rat cells (e.g., combination of SV40 large T antigen and oncogenic HRAS^G12V^) also rendered naked mole-rat cells malignant in the laboratory. The fact that naked mole-rats are resistant to carcinogenesis but their cells in the laboratory are not implicates the microenvironment in the prevention of cancer. One wonders whether it is possible to show that an embryonic or stem-like microenvironment in the naked mole-rat somehow makes it a Methuselah impervious to cancer.

In fact, such classic experiments have already been done. In 1974, Damjanov and Solter [10] demonstrated that normal stem cells derived from the gonadal ridge converted into embryonal carcinoma when grafted onto an adult mouse testis. Conversely, Illmensee [11] showed that embryonal carcinoma cells inserted into the inner cell mass of a mouse blastocyst (implantation-stage embryo) behaved like normal stem cells and became part of a normal mosaic mouse. In addition, mouse B16 melanoma cells failed to form tumors when exposed to embryonic skin in utero [12], and human metastatic melanoma cells were not tumorigenic when implanted in a chicken or zebrafish embryo [13,14].

## 4. Down’s Syndrome

Down’s syndrome is a genetic disorder characterized by the presence of an extra chromosome 21. Patients have developmental delays, cognitive impairment, heart defects, muscle abnormality, neurological anomalies, et cetera.

Despite the multiple clinical ailments patients with Down’s syndrome endure, the genetic disorder somehow protects them from another devastating affliction, namely cancer.

Even though Down’s syndrome is associated with increased risks of numerous cancer-promoting factors, including chromosomal instability, increased DNA damage, inflammatory conditions, immune abnormalities/deficiency, and excessive weight gain, patients with Down’s syndrome somehow have a lower risk of developing solid tumors [15].

Galat et al. [16] demonstrated that the impairment of mesodermal progenitor functions, as reflected in the downregulation of vital processes involving the cytoskeleton, angiogenesis, cell cycle, and extracellular matrix organization, affected endothelial cells’ response to external stimuli, cell migration, and immune response and contributed to the resistance to solid cancer formation in Down’s syndrome.

Intriguingly, they also discovered that although cancer-related genes on chromosome 21 were upregulated, they were downregulated on all other chromosomes, suggesting that trisomy 21 induced modified regulatory and genome-wide compensatory mechanisms that engendered the tumor-resistant phenotype.

It would be fortuitous if our attempts to explain the reduced risk of solid tumors associated with Down’s syndrome led to the discovery of better ways to prevent cancer and improvements in our ability to treat, if not cure it [17].

## 5. Senescence

Evidently, it is not just the genes on chromosome 21 that are upregulated in Down’s syndrome (i.e., “gene dosage effect”), but gene expression across every chromosome has run amok.

Meharena et al. [18] detected structural DNA changes and gene expression disruptions compatible with senescence in neural progenitor cells derived from patients with Down’s syndrome. They identified global genetic changes in all chromosomes, not just local genetic defects in chromosome 21. Importantly, they found that cellular context mattered: abnormal senescence occurred in neural progenitor cells, but not in stem cells.

Notably, people with Down’s syndrome age faster than other people. They have more senescent cells and impaired immune systems that do not eliminate the accumulating senescent cells.

One burning question pertaining to this peculiar observation: do patients with Down’s syndrome have fewer stem cells? Assuming cancer has a stem-cell origin, we postulate that this could be the reason they age faster and why they are at a lower risk to develop cancer [19,20].

It is of interest that certain tumor suppressor genes, such as p16^INK4a^, prevent uncontrolled cellular proliferation. Consequently, impaired p16^INK4a^ does not induce cell cycle arrest and cannot defend against runaway cancer growth. Conversely, overactive p16^INK4a^ causes permanent cell cycle arrest and senescence. Therefore, a negative end result of p16^INK4a^ overdrive is limited self-renewal and restricted stem cell proliferation, which promote senescence and accelerate aging [21,22,23].

Similarly, telomere shortening is a potent tumor suppressor—it may prevent cancer but lead to senescence [24,25]. In many respects, DNA repair is also a potent tumor suppressor and serves the same function as telomere shortening. Again, DNA damage triggers DNA repair, which imposes cell cycle arrest and induces senescence in order to prevent cancer [26].

An individual may be protected from the risks of cancer by an active cadre of tumor suppressor genes as one grows older, but the person becomes increasingly vulnerable to stem-cell senescence and is at the mercy of the aging process. Although increased senescence decreases the chance of tumor formation, we may not experience any survival advantage because of accelerated aging [27].

After all, oncogenes wax, whereas tumor suppressor genes wane during fetal development. However, the opposite is true during the aging process. Therefore, the benefit of longevity needs to be balanced against the risk of malignancy. Perhaps how we manage to conserve stemness and delay senescence is key [19,20,28].

## 6. Progeria

Interestingly, cancer is also less prevalent in patients with progeria. Perhaps there is a common denominator between Down’s syndrome and progeria with regard to the aging process and malignant formation.

We speculate that in both scenarios, depleted stem cells lead to less cancer, and that defective progenitor cells result in the clinical sequelae of Down’s syndrome and progeria.

If cancer is somehow related to the aging process, then in an abbreviated version of the aging process, as occurs in patients with progeria, we should observe accelerated cancer formation. But the opposite is true. Patients with Hutchinson-Gilford progeria syndrome (HGPS) or “childhood progeria” die from old age, without cancer.

HGPS is a rare genetic disease associated with accelerated aging in certain organ and tissue sites. Children afflicted by HGPS develop generalized cardiovascular disease. They often die from a heart attack or stroke when they are 13 years old. They suffer from hair loss, wrinkled skin, and stiff joints. Interestingly, HGPS patients do not experience metabolic, endocrine, or immune dysfunction, and they do not develop cataracts, diabetes, or hyperlipidemia. They have normal intelligence and emotions.

Importantly, cellular context versus genetic content has profound therapeutic implications for both HGPS and cancer [29,30,31,32]. Halaschek-Wiener et al. [33] suggested that progerin, a truncated form of the nuclear structural protein lamin A, caused increased apoptotic cell death and exhausted tissue-specific stem cell regeneration in HGPS. Tissues that require constant growth (hair and skin) and repair (joints and blood vessels) are more affected than those that do not (eye lenses, brain or immune cells, and the endocrine system). Therefore, a depleted stem-cell pool in HGPS may avert the development of cancer at its cellular, if not genetic, origin. After all, all HGPS cells have the same genetic defect, but not all cells are created equal. It is the progenitor cells rather than the progeny cells that determine regeneration vs degeneration, benignity vs malignancy.

When it concerns stem-cell disease and stem-cell depletion, Werner syndrome or “adult progeria” provides an unexpected control. Unlike patients with HGPS, patients with Werner syndrome are diagnosed (age 24 years) and die (54 years) later in life. They suffer from cataracts, diabetes, and cancer. Mutations in the WRN gene that cause Werner syndrome affect helicase, which is important for maintaining DNA and telomere stability. Because the stem-cell pool in Werner syndrome is either not depleted or delayed in its depletion, we suspect that cancer formation is either not affected or less affected compared with that in HGPS [20].

## 7. Stemness

In many respects, stemness accounts for and accommodates all the conventional hallmarks of cancer, including autonomy, metastasis, dormancy, heterogeneity, immune evasion, and genetic instability [19,20]. When stemness is a preeminent feature and predominant property of cancer, it alludes to a stem-cell origin and a stem-like nature of cancer [34,35,36,37].

Consequently, oncology recapitulates ontogeny. Carcinogenesis encapsulates embryogenesis. Embryonic, fetal, and stemness biomarkers and pathways continue to resurface and unfold in cancer research and cancer care. They turn out to be relevant tests and reliable targets for the diagnosis, prognosis, and therapy of cancer. When cancer is stemness going awry, rogue, or berserk, perhaps the best way to beat cancer is to know how to fix, tame, or calm stemness in some way or another.

It is well known that when the heart is damaged, as from a heart attack, it does not regenerate from the damage nor repair the damage very well. One possible reason for this deficiency is that the heart has very few or no stem cells to regenerate the organ or to repair the tissue. After all, this is what stem cells are supposed to do: regenerate and repair.

Indeed, Kretzschmar et al. [38] searched but did not find any evidence of stem cells being present in the heart at all. Perhaps this should not be a surprise, given the observation mentioned above that the heart does not regenerate and repair very well when damaged.

The surprise is with an organ such as the liver, which regenerates and repairs only too well. What is unique with the liver is that progeny differentiated cells may behave like progenitor stem cells. When differentiated liver cells have stemness properties, it does complicate the definition of stemness, and it does confuse the meaning of stemness. Again, a critical question is whether stemness is reversible and fluid, or whether it is merely unequal and there is a hidden hierarchy of increased stemness down a ladder of decreased stemness in the liver cells.

As a matter of fact, Wang et al. [39] have discovered that some liver cells are more pluripotent (more stemness) than others. They demonstrated that those liver stem cells have the capacity to differentiate into diverse cellular lineages and are clustered around the liver’s central veins. Their finding suggests that a stem cell hierarchy is also present in the liver.

Therefore, the difference between the liver and the heart may be in the quantity and quality of stemness in a continuum rather than a dichotomy of cells. There are relatively more cells with more stemness in the liver, which ensure and enable it to regenerate and repair at a greater capacity than those in the heart.

In many respects, this robust stemness capability is at play in almost all fetal organs and tissues, not just in the liver. Perhaps the adult liver is one of the few organs that happens to retain more fetal characteristics, including more stemness reserves, than all other organs in our body. One would expect that all the organs in a fetus will regenerate and repair much better than those in an adult, including the liver and perhaps even the heart (before the heart stem cells become depleted). This is the essence of stemness and aging.

## 8. Methylmalonic Acid

A correct theory about the stem cell vs genetic origin of cancer may help us ascertain whether methylmalonic acid (MMA) is the driver or passenger in a runaway cancer. Perhaps it will enable us to evaluate whether MMA is the actual engine or just the chassis of malignancy.

Gomes et al. [40] found that MMA levels are higher in the blood of older compared with younger people. MMA activates TGF-beta1, SOX4, and epithelial-mesenchymal transition (EMT) and endows cancer cells with more aggressive behaviors, such as metastatic potential and drug resistance, presumably to a greater degree in older people than in younger people.

Except that many cancers happen to be more aggressive and deadly in younger than in older patients [41,42,43]. Furthermore, MMA, TGF-beta1, SOX4, and EMT may also play important roles and perform vital functions under normal circumstances in normal stem cells of both young and old people without cancer.

Because older people are more likely to be depleted of stem cells, perhaps they need more MMA, TGF-beta1, SOX4, and EMT to help them preserve their remaining stem cells so that they can stay healthy, regenerate tissues, repair wounds, et cetera. In normal stem cells, this MMA/TGF-beta1/SOX4/EMT axis is essential for our good health. In cancer stem cells, this same axis makes cancer more vicious and is detrimental to our health.

According to the scientific method, a different theory and hypothesis will give us a vastly different perspective about cancer, guide us to design different experiments to interrogate the theory and investigate the hypothesis, and influence our interpretation of the results of those experiments.

Therefore, the questions we pose about MMA and how we answer those questions within the context of a stem-cell theory of cancer and aging matter. For example: Why would an anti-aging molecule, MMA, which induces a stemness gene, SOX4, and the EMT phenotype, be present in higher levels in older people? Could it be because there are fewer stem cells present? Is it because in older people the regenerative potential of stemness pathways is less robust and we need higher levels of MMA to keep up stemness? Is MMA related to the efficiency or activity of metabolism of older people—TCA vs glycolysis? Is MMA related to a general lower activity of tumor suppressors in older vs younger people? Is MMA an effect rather than the cause of aging, like wrinkles and gray hair?

Suppose we inhibit SOX4 in those experiments so that the mice will have more attenuated EMT and less aggressive tumors. They may end up having fewer normal stem cells and accelerated aging. However, those mice may not live longer, because they will die from aging, if not from cancer.

This is the irony of reductionist scientific research. Often enough, we peer at a small piece or imagine a figment of the big picture, but do not actually see or envision the whole. Importantly, we need to be aware that when we have a false theory and the wrong hypothesis, we may have a distorted view of reality and aging or an erroneous perspective of the origin and nature of cancer.

## 9. Microbiome

A stem-cell theory of cancer predicates that not only does the cell affect the niche, the niche also affects the cell. It turns out that the microenvironment reaches beyond the neighborhood. It also extends into a whole society of cells, including the microbiome.

Cabreiro et al. [44] found that metformin—a drug used to treat elevated blood sugar and metabolic syndrome in type 2 diabetic patients—also elicits anti-aging effects. Specifically, metformin prolongs the lives of nematode worms (*C. elegans*) but only if the worms have microbes in their guts. The microbiome also seems to affect the longevity of killifish—old killifish live longer if they consume the excrement of young ones, suggesting that either old microbiomes shorten their lifespan or young microbiomes lengthen it [45].

Han et al. [46] demonstrated that nematodes that lived longer than their counterparts did so because of the presence of a microbe (*E. coli*) in their gut, rather than because of their genetic makeup (i.e., they are genetically identical). They discovered that out of about 4000 strains of *E. coli*, 29 extended the nematode’s lifespan by at least 10%. Interestingly, 19 of those 29 strains also prevented age-associated diseases, including cancer.

Intriguingly, what enhanced *C. elegans*’ quantity and quality of life could be deduced from two *E. coli* strains with a missing gene that produces colanic acid, which coats the cell surface of many gut microbes. Without the gene, the microbes make copious amounts of colanic acid and the nematodes live longer. Without colanic acid, they no longer benefit from extended lives.

It is plausible that colanic acid prolongs the nematodes’ lives by enriching their mitochondria and empowering them to better utilize energy and withstand stress. However, colanic acid may only be an enticing clue rather than the definitive explanation of the power of a healthy microbiome that promises a longer and better life.

## 10. Diet and Exercise

When we aim for a healthy microbiome and crave longevity and quality of life, perhaps we need to look no further than a healthy diet and active lifestyle. After all, when we examine the habits of super centenarians, a healthy diet and active lifestyle do appear to be part of their secret to a long and healthy life [47,48,49]. They also seem to be spared many age-related or -unrelated maladies, including cancer.

It may not be a coincidence that the aryl hydrocarbon receptor (AHR) in the gut modulates our microbiome and mediates our longevity. AHR is a regulator of cytochrome P450, which metabolizes xenobiotic chemicals coming through our alimentary canal. Importantly, AHR displays stemness properties and dispenses stem-like functions because it regulates metabolic enzymes, moderates immunity, and balances stem cell pluripotency vs differentiation [50,51]. After all, AHR binds exogenous ligands, such as natural plant flavonoids, polyphenols, and indoles, that we ingest to feed ourselves, as well as to feed the microbiome in our guts.

*Akkermansia muciniphila* is one of those microbes in our gut that thrives on natural plant flavonoids, polyphenols, and indoles. Apparently, it protects us from metabolic syndrome [52] and malignancy [53,54]. It may also assure our longevity. *A. muciniphila*, as its name implies, produces the mucin that lines our gut. This mucus layer prevents the leaking of undigested food particles and bacteria into the blood stream, which initiates an immune and inflammatory response and instigates autoimmunity and malignancy, respectively [55]. Notably, *A. muciniphila* is associated with improved clinical outcomes in patients with lung and kidney cancers [53]. It seems to enhance the anticancer effects of immunotherapy [53,56,57].

In addition, Spencer et al. [58] reported that advanced melanoma patients who consumed at least 20 g of dietary fiber per day lived longer with anti-PD1 therapy. They found that dietary fiber increased the presence of a family of bacteria, Ruminococcaceae, in the gut flora and the production of certain short-chain fatty acids, such as propionate, which provided salubrious antitumor effects. Probiotics, on the other hand, induced a weakened immune response.

## 11. MicroRNAs

Indeed, research on microRNAs (miRNAs) suggests that taking good care of our stem cells may be good for our antiaging and anticancer strategies. After all, normal and cancer stem cells share a similar miRNA profile. A stem-cell theory of cancer suggests that what keeps normal stem cells healthy may also keep cancer stem cells restrained through the modulation of certain miRNAs.

Humans have about 25,000 genes and more than 400 miRNAs. A miRNA regulates gene expression by base-pairing with target mRNAs and inhibiting their expression. Each miRNA can target many genes (dozens to hundreds). As many as 30% of all genes are thought to be under miRNA control. Among the many epigenetic modifiers, miRNA is both versatile and specific.

Utikal et al. [59] distinguished miRNAs that are exclusively expressed in embryonic stem cells (ESCs) but not in their differentiated progeny. For example, miR145 posttranscriptionally diminishes the expression of OCT4, KLF4, and SOX2, thereby suppressing hESC self-renewal and driving differentiation towards mesodermal and ectodermal lineages. Similarly, let-7 negatively regulates the levels of pluripotency-associated proteins Sall4, myc, and Lin28, whereas miR-294 induces their expression. miR-223 induces differentiation of hESCs by targeting IGF-1R, which may play a role in the maintenance of pluripotency in the IGF/Akt pathway.

In addition, miR-200s regulate a well-known stemness phenotype, namely EMT, by controlling the epithelial state, stem-like properties, and therapeutic response [60]. Furthermore, the miR-34 family has been shown to reduce the protein levels of important pluripotency-associated factors such as Nanog, Sox2, and N-myc [61]. However, miR34a is also known to be a powerful suppressor of prostate cancer stem cells and metastasis by directly inhibiting CD44 [62]. Interestingly, miRNAs derived from cancer cells (and likely also stem cells) can alter the molecular profile of neighboring cells and attune the microenvironment through exosomes [63].

Importantly, certain food products contain natural epigenetic modifiers that regulate various miRNAs, modulate stemness activity, and provide salubrious antiaging functions as well as anticancer effects [64,65,66,67]. For example, fermentable fiber/pectin (butyrate) and fish oil (DHA and EPA) increased the expression of miR-19b, miR-26b, miR-27b, miR-200c, and miR-203 and decreased the expression of their predicted targets, some of which have been shown to mediate oncogenic signaling [68]. Curcumin reduced EZH2 expression and increased a panel of tumor suppressive miRNAs comprising miR-26a, miR-101, miR-146a, miR-200b, and miR-200c [69]. Genistein up-regulated miR-200 and let-7, which was associated with the down regulation of ZEB1, slug, and vimentin, known to play a role in EMT [70]. 

It is conceivable that as we gain more knowledge and understanding about a stem-cell origin and nature of aging and of cancer, we will know better how to help ourselves by taking advantage of dietary habits and lifestyle changes that delay aging and prevent cancer.

## 12. Proteostasis

It is evident that stem cells are adept in protein recycling. Good house cleaning keeps stem cells fresh and pristine, healthy and youthful. It is essential that we regularly remove the garbage in the house and routinely replace the broken parts in our car. It is like a deciduous tree shedding its old leaves every fall and sprouting new leaves every spring. Like DNA repair, protein recycling keeps stem cells vibrant, robust, and ageless.

Bohnert et al. [71] demonstrated that V-ATPase, a lysosomal proton pump, triggers proteostasis and removes damaged proteins to ensure longevity in stem cells. Indeed, one of the most amazing stem cells in the body, namely oocytes, underwent highly efficient house cleaning of protein aggregates before fertilization in *C. elegans* and under the influence of progesterone in the frog *Xenopus laevis*. It was as if the oocytes needed to ensure a clean slate for the next generation or the next cycle of life. They accomplished this task by employing a primordial process, namely autophagy, and using lysosomes to recycle and reclaim old proteins.

If autophagy and protein recycling affect stem cells and play a central role in rejuvenation, can we harness their power and potential to fulfill our antiaging and anticancer goals?

If stem cells have an innate capacity to regenerate by keeping DNA immaculate and proteins impeccable, perhaps we can support those functions in our daily lives. We may be able to take advantage of good habits, such as consuming nutritional food and adopting an active lifestyle—utilize the same tools that super centenarians do to make sure that our stem cells remain healthy and strong.

However, when we do the opposite and eat junk food, smoke cigarettes, or drink alcohol, we pay the consequences. When we keep our stem cells on their toes with incessant stress and unremitting damages, we will exhaust our stem cell reserve and accelerate the aging process. We will overreach and overcome their checks and balances and banish the hapless stem cells into the purgatory of malignancy.

## 13. Conclusions

A stem-cell theory of aging and cancer reiterates a fundamental oncological principle: although genetic makeup may be pivotal, cellular context is paramount. When the genome and epigenome that regulate aging and malignancy are also stemness genes and stem-like properties, they reaffirm the key role that stem-cell quality and quantity play in longevity and cancer (Figure 1).

We suspect that long-lived, cancer-spared mammals maintain a youthful genome and epigenome because they are equipped with a larger and healthier pool of stem cells. We contemplate that people with Down’s syndrome and progeria age faster but have fewer cancers because they are depleted of stem cells and therefore have fewer opportunities for stem-cell defects that render one prone to cancer formation.

It seems that these extraordinary experiments of nature provide tantalizing evidence, if not definitive proof, that cancer is a stem-cell disease—fewer aberrant stem cells, fewer cancers; no defective stem cells, no cancer.

Therefore, the benefit of longevity needs to be balanced against the risk of malignancy. Intuitively, how we manage to conserve stemness and delay senescence is key. Empirically, good habits and an active lifestyle, such as a healthy diet and moderate exercise, may do. After all, that is what the *C. elegans* and *X. laevis* do to ensure longevity and quality of their ova, too.

Perhaps we will find solace in the wisdom of Chevalier that old age isn’t so bad if we consider that the alternative is death—and maybe cancer.

## Figures and Tables

**Figure 1 cancers-14-01338-f001:**
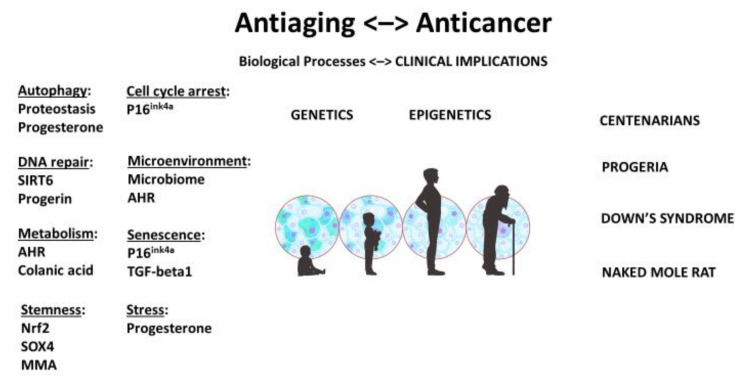
Unified theory and stem-cell origin of cancer and aging: biological processes and clinical implications. As we age, stem-cell pool becomes depleted and stem cells may be damaged. AHR: aryl hydrocarbon receptor; MMA: methylmalonic acid. Illustration by Benjamin Tu.

## Data Availability

The data presented are available in the references cited in this article.

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
