# Peer review of "Stem-Cell Theory of Cancer: Implications for Antiaging and Anticancer Strategies"

_cancers, 2022, doi:10.3390/cancers14051338_

Round 1

Reviewer 1 Report

The MS entitled "Stem Cell Theory of Cancer: Implications for Antiaging and Anticancer Strategies" is a well-written and attractively presented "Perspective."

I request authors to consider discussing the following relevant articles in different sections:

Section 3. Methuselah: Please provide a one-line definition of the word "Methuselah" and connect its perspective in the context of stem cell origin of aging and cancer.

Here are a few articles to consider for the discussion:

  • In search of Methuselah: estimating the upper limits to human longevity, Science.

1990 Nov 2;250(4981):634-40. doi: 10.1126/science.2237414.

Section 4. Down's Syndrome

Here are a few articles to consider for the discussion:

  • Int J Mol Sci. 2017 Jun 7;18(6):1218. doi: 10.3390/ijms18061218. Down's Syndrome and Triple-Negative Breast Cancer: A Rare Occurrence of Distinctive Clinical Relationship

Section 5. Aging and Cancer

The MS contains "2. Age and Cancer" and "5. Aging and Cancer". It is confusing. Please streamline it.

Here are a few articles to consider for the discussion:

  • Nat Rev Genet. 2019 May;20(5):299-309. doi: 10.1038/s41576-019-0099-1. Telomeres and telomerase: three decades of progress
  • N Engl J Med. 2009 Oct 8;361(15):1475-85. doi: 10.1056/NEJMra0804615. DNA damage, aging, and cancer
  • N Engl J Med. 2009 Oct 8;361(15):1475-85. doi: 10.1056/NEJMra0804615. DNA damage, aging, and cancer

Section 6. Progeria

Here are a few articles to consider for the discussion:

  • Expert Opin Investig Drugs. 2012 Jul;21(7):1043-55. doi: 10.1517/13543784.2012.688950. Epub 2012 May 24. Lonafarnib for cancer and progeria
  • Nat Med. 2019 Mar;25(3):423-426. doi: 10.1038/s41591-018-0338-6. Epub 2019 Feb 18. Development of a CRISPR/Cas9-based therapy for Hutchinson-Gilford progeria syndrome
  • Nat Med. 2021 Mar;27(3):526-535. doi: 10.1038/s41591-021-01262-4. Epub 2021 Mar 11. Systematic screening identifies therapeutic antisense oligonucleotides for Hutchinson-Gilford progeria syndrome.
  • 2021 Jan 21;184(2):293. doi: 10.1016/j.cell.2020.12.029. Farnesyltransferase inhibition in HGPS

Section Stemness: Will it be nice for the internal consistency to add the section under the subheading stemness as a section like "7. Stemness"?

Here are a few articles to consider for the discussion:

  • Stem cells, cancer, and cancer stem cells. Reya T, Morrison SJ, Clarke MF, Weissman IL. Nature. 2001 Nov 1;414(6859):105-11. doi: 10.1038/35102167.
  • Cancer stem cells: nature versus nurture. Korkaya H, et al. Nat Cell Biol. 2010. PMID: 20418873.
  • Roots and stems: stem cells in cancer; Kornelia Polyak, William C Hahn; Nat. Med. 2006 Mar;12(3):296-300. doi: 10.1038/nm1379.
  • Stem Cell Res Ther. 2013 Mar 8;4(2):21. doi: 10.1186/scrt169. The 3R principle: advancing clinical application of human pluripotent stem cells

Since wnt signaling is implicated in development, self-renewal as well as tumorigenesis, and the canonical Wnt cascade has emerged as a critical regulator of stem cells; authors can discuss the wnt pathway in the context (Wnt signalling in stem cells and cancer. Reya T, Clevers H. Nature. 2005 Apr 14;434(7035):843-50. doi: 10.1038/nature03319. PMID: 15829953).

Since the MS aims to present a comprehensive perspective of stem cell origin of aging and cancer, a separate section on miRNA and aging in the context of epigenetics and cancer will be a treat for the readers.

Here are a few articles to consider for the discussion:

  • Effects of resveratrol, curcumin, berberine and other nutraceuticals on aging, cancer development, cancer stem cells, and microRNAs, Aging (Albany NY). 2017 Jun 12;9(6):1477-1536. doi: 10.18632/aging.101250.
  • MicroRNAs and Epigenetics Strategies to Reverse Breast Cancer, Cells. 2019 Oct 8;8(10):1214. doi: 10.3390/cells8101214.
  • miRNA-Processing Gene Methylation and Cancer Risk, Cancer Epidemiol Biomarkers Prev. 2018 May; 27(5):550-557. doi: 10.1158/1055-9965.EPI-17-0849. Epub 2018 Feb 23.
  • Human Aging and Cancer: Role of miRNA in Tumor Microenvironment, Adv Exp Med Biol. 2018;1056:137-152. doi: 10.1007/978-3-319-74470-4_9.

The discussion including the above articles will also throw light on the "Anticancer Strategies" of the article.

Minor comments:

Authors state that "Funding: This research received no external funding. Yet authors acknowledge that "This work was supported in part by the National Institutes of Health through UT MD Anderson's Cancer Center Support Grant CA016672 and by donations from the Realan Foundation and  Mr. Harendra Mankodi (S.T.)." It is confusing to the reader. Please clearly state the funding and support information.

Reviewer 2 Report

The authors of "Stem Cell Theory of Cancer: Implications for Antiaging and Anticancer Strategies" present a very interesting topic from an interesting point of view. But for the most part, the article seems like a collection of information without an in-depth analysis of the concepts. A more detailed presentation of the data is necessary in order to highlight the scientific importance and the clinical significance of the proposed approach.
Specific points:
1) For example in the paragraph "Importantly, 81 longer-lived mammals have more stable epigenetic marks, (...), ie, with 83 stemness marks." there is no reference of any specific gene(s). 

2)The paragraph "Aging and Cancer" is a characteristic example of more details needed. More than that, in the paragraphs "It is of interest that certain tumor ..(...) because of accelerated aging [22]." the p16 gene is mentioned but no connection with the notion aging and cancer is made. It seems that these two paragraphs are presenting information and forgetting to make a point! 

3) The opposite is happening in the paragraph "Stemness" where details are presented about methylmalonic acid (MMA) dependent signaling and the general concept is missing. The same more or less applies to the paragraph "Microbiome". 

4. In the paragraph "Antiaging and Anticancer" one fails to understand why this was picked as an appropriate title. Maybe "Proteostasis" or something similar is a better choice. Also, the paragraph "Bohnert et al. [44] demonstrated t... frog Xenopus laevis." needs to be rewritten in order to be understood. 

5. Figure 1 is presented too early. Meaning the readers fail to understand it since most of the elements of the figure haven't been described anywhere. So, I propose another figure to be added in the begining of the manuscript where the authors can graphically present their core idea and figure 1 to move further down in the text after all mentioned molecules/mechanisms have been discussed. Furthermore a more detailed figure legend is needed for figure 1. Also what does "Illustration by Benjamin Tu" means? The sourse of the picture needs to be properly cited. 
